# Application of Organic Fertilizer Changes the Rhizosphere Microbial Communities of a Gramineous Grass on Qinghai–Tibet Plateau

**DOI:** 10.3390/microorganisms10061148

**Published:** 2022-06-02

**Authors:** Kun Ma, Yingcheng Wang, Xin Jin, Yangan Zhao, Huilin Yan, Haijuan Zhang, Xueli Zhou, Guangxin Lu, Ye Deng

**Affiliations:** 1Collage of Agriculture and Animal Husbandry, Qinghai University, Xining 810016, China; m18294258973@163.com (K.M.); yingcheng_w@163.com (Y.W.); 18894310895@163.com (X.J.); 18893093541@163.com (Y.Z.); yan06112021@163.com (H.Y.); 15297196583@163.com (H.Z.); zhouxuelia@163.com (X.Z.); 2Experimental Station of Grassland Improvement in Qinghai Province, Gonghe 813000, Qinghai, China; 3Research Center for Eco-Environmental Sciences, Chinese Academy of Sciences, Beijing 100085, China; 4College of Resources and Environment, University of Chinese Academy of Sciences, Beijing 100190, China

**Keywords:** organic fertilizer, rhizosphere microbial communities, Qinghai–Tibet Plateau, microbial assembly, arbuscular mycorrhizal fungi

## Abstract

The effects of organic fertilizer application on the soil microbial community in grassland systems have been extensively studied. However, the effects of organic fertilizers on the structure of rhizosphere microbial communities are still limited. In this study, the diversity and composition of rhizosphere microbial communities of a gramineous grass *Elymus nutans* under organic fertilizer treatment were studied in an artificial pasture on Qinghai–Tibet Plateau. After a growing season, the application of organic fertilizer not only increased the height and biomass of *Elymus nutans*, but also changed the rhizosphere microbial compositions. In particular, organic fertilizer increased the diversity of rhizosphere bacterial community and inhibited the growth of pathogenic bacteria such as *Acinetobacter*, but the opposite trend was observed for the diversity of fungal community. The assembly process of fungal community was changed from a stochastic process to a deterministic process, indicating that selection was strengthened. Additionally, both the infection rate of arbuscular mycorrhizal fungi (AMF) toward host plants and the development of AMF-related structures were significantly increased after the application of organic fertilizer. Our study demonstrated that the addition of organic fertilizer to artificial pasture could improve the growth of grass through the alteration of the rhizosphere microbial communities. Organic fertilizer had a greater selectivity for the bacterial and the fungal communities that enhanced the niche filtration in this community, further benefiting the yield of forages.

## 1. Introduction

The Qinghai–Tibet Plateau is an important terrestrial ecosystem with large areas of grazing alpine grassland but is a particularly fragile ecological region [1,2]. In recent years, large-scale overgrazing has caused a series of environmental problems, such that the establishment of artificial pastures became a widely applied pattern for improving forage productivity [3,4,5]. Many studies have shown that planting annual forages with fast growth and high yields can solve the problem of forage shortage and improve the quality of forage in alpine grassland [6]. Many studies attempted to replant leguminous forage in alpine grassland, but the cold climate made it difficult to overwinter. It was later found that alpine grassland has a quick growth of gramineous forage [7]. Screening high-protein gramineous forage could also improve the productivity of alpine grassland. Conversely, the consumption of soil nutrients also gradually increases. Therefore, the application of fertilizer is often used to improve the yield and quality of forage in alpine grasslands [8].

The dominant role of microorganisms in soil nutrient cycling processes affects plant diversity and stability in grasslands [9,10]. The microbes in the rhizosphere play important roles in terrestrial ecosystems and are sensitive indicators of soil quality, productivity, and sustainability [11]. They also have important roles in plant nutrient uptake and resistance to adverse conditions [12,13], both indirectly and directly, through symbiotic relationships (e.g., mycorrhizal) with plants or by producing phytohormones to promote plant growth [14]. Many studies have found that organic material application is beneficial in promoting the soil organic matter contents and bacteria growth, which are important to further increase the effectiveness of soil nutrients, improve soil fertility, and enhance soil productivity [15,16]. Meanwhile, through network analysis, Ji et al. [17] found that long-term application of organic fertilizers resulted in the formation of potential functional groups, as well as interactions between soil microorganisms, likely due to the enhanced soil stability and buffering capacity of organic fertilizers. Compared to bacteria, fungi are more sensitive to exogenous organic fertilizers. It was noted that nutrients in organic fertilizers are more selective for fungal taxa, thus enhancing the effect of ecological niche filtering on fungal communities, and that organic fertilizer application can promote species extinction, thus reducing fungal abundance. Fertilizer incorporation affects soil fungal diversity and community assemblages mainly by changing the soil ecotone width and available substrate types in the soils [18,19].

Overall, two ecological theories are proposed to explain the spatial patterns in macro- and micro-ecology [4]. The niche-based theory and the neutral-based theory constitute two important and complementary mechanisms for understanding microbial community assembly [20]. The niche-based process assumes that microbial communities are determined by deterministic abiotic factors and biotic factors, attributed to different habitat preferences and adaptive capabilities of microorganisms. Neutral theory, on the other hand, assumes that stochastic processes, such as birth, death, migration, species formation, and limited dispersal, shape microbial community structure, and it postulates that microorganisms present a stochastic balance between losses and gains in taxa [21]. The normalized stochasticity ratio (NST) reflects the contribution of stochastic assembly relative to deterministic assembly, as a function of the magnitude of the difference between observed and null expectations, as a quantitative measure of stochasticity. The relative importance of deterministic and stochastic processes in different community assemblies is quantified by comparing the numerical distribution of normalized stochasticity ratio between sample pairs within different groups. The authors of [20] confirmed, by simulating communities under a variety of conditions such as abiotic disturbance, competition, environmental selection, and spatial scale, that, while NST has limited performance at large spatial scales or under higher environmental noise, it shows high accuracy in other scenarios. Chen et al. [22] conducted a large-scale survey in northern Australia to explore the mechanisms of microbial community assembly in anthills and found that deterministic processes exhibited a significantly weaker impact on bacteria (NST = 45.23%) than on fungi (NST = 33.72%). Wang et al. [4] used NST to analyze potential models of soil microbial community assembly in degraded grasslands of the Tibetan Plateau region, and they found that grassland degradation caused a shift from stochastic to deterministic processes in fungal community microbial assembly. Although stochastic processes are thought to play an important role in shaping microbial community structure, the importance of ecological stochasticity in influencing community structure was previously often overlooked due to the difficulty of defining stochasticity and the lack of methods used to represent it. With the rapid development of microbial ecology, many conceptual models of microbial community formation have been proposed. Community diversity and distribution patterns can provide evidence for the process of community assembly. For a broad range of prokaryotic communities, the relative abundance and frequency of different taxa observed in a sample can be explained by the neutral community model (NCM), a stochastic, birth–death immigration process that does not explicitly represent determinants and, therefore, cannot fully or literally describe community construction (assembly). However, it successfully demonstrates that chance and migration are important forces in shaping prokaryotic community patterns. Gad et al. [23], using an 18S rRNA high-throughput sequencing approach, applied NCM analysis to reveal the seasonal succession pattern of planktonic eukaryotic microorganisms, which is strongly driven by stochastic processes. Inspired by this, we combined NST and NCM analysis methods to explore the effects of organic fertilizer treatments on microbial community assembly patterns.

The role of AMFs (arbuscular mycorrhizal fungi) in agricultural production is controversial, but they are undoubtedly an important component of subsurface ecosystems. However, high inputs of fertilizers, phosphorus, and fungicides may inhibit the formation of AMFs and prevent their beneficial contribution to crop nutrition and health [24]. A recent study tested AMF diversity in 26 cropland sites and found that the average amount of AMFs in organic soils was much higher than that in conventional agricultural soils, evidence that organic management of agroecosystems helps restore and maintain soil biodiversity [25]. However, some studies have found that higher application rates of organic amendments can lead to large excesses of N and P, which may reduce the diversity of native AMF communities [26]. It is widely recognized that soil biodiversity is a key determinant of above-ground biodiversity and function in terrestrial ecosystems. In general, the diversity of AMF determines the function of its ecosystem; a higher diversity of AM fungi reflects greater importance for improving crop yield, nutrients, biodiversity, and ecosystem stability [27]. In addition, it has also been shown that the link between plants and soil microorganisms can be modified by altering the symbiotic relationship between clumping mycorrhiza fungi (AMF) and plants [28].

In general, the effects of organic fertilizers on rhizosphere microorganisms have been widely studied in the agro-ecosystem in Asia, but are very rarely reported at high altitudes, especially in high-altitude grasslands. The application of organic fertilizer can promote the conversion of soil residual nitrogen to effective acidotic nitrogen, so as to improve soil fertility. Short-term application of organic fertilizer can significantly affect the pH of soil total nitrogen and organic matter, while organic fertilizer can improve soil physical structure, increase soil fertility, and improve the abundance and diversity of soil microbiome and microbial community structure [29,30,31]. In order to clarify the effects of organic fertilizers on the structure and diversity of rhizosphere microbial communities of forage grasses on the Qinghai–Tibetan plateau, we conducted control and organic fertilizer application experiments in artificial grass of *Elymus nutans*, and several null model methods were used to measure the microbial assembly processes during organic fertilizer application. We hypothesized that (i) organic fertilizer could significantly alter the diversity and community structures for both bacterial and fungal communities of the rhizosphere, and (ii) organic fertilizers could strengthen the selection of microbial assembly of the rhizosphere.

## 2. Methods and Materials

### 2.1. Study Sites and Organic Fertilizer Treatment

Qinghai Province Grassland Improvement Experiment Station is located on the western shore of Qinghai Lake (99°35′ E, 37°05′ N; 3270 m). The area has an alpine plateau climate with sufficient light, strong UV irradiation, cold winters and cool summers, little drought, and high winds, and it is representative of the alpine grassland ecological environment on the Qinghai–Tibet Plateau. According to the data provided by the nearby weather station, we found that the annual average temperature is −0.7 °C, the average temperature of the hottest month (July) is 17.5 °C, the average temperature of the coldest month (January) is −22.6 °C, and the extreme temperature is −34.3 °C. The average frost-free period is 78.7 days, with no absolute frost-free period. The number of sunshine hours is 2670 h, the annual cumulative temperature ≥0 °C is 1331.3 °C, the annual precipitation is 368.11 mm, the annual evaporation is 1495.3 mm, and the relative humidity is 58%.

The experiment was conducted in May 2020, with a 200 m^2^ plot selected as the experimental area for planting introduced grass forage at the forage experiment station of the Qinghai Provincial (Figure 1) Grassland Improvement Experiment Station. The experiment was divided into fertilized and unfertilized plots, each replicated six times for a total of 12 plots. Each plot had an area of 15 m^2^ (3 m × 5 m), with a spacing of 1 m between plots, and they were sown with grass forage in strips at 20 cm intervals in furrows 5 cm deep, with a total of 12 rows sown per plot. The plots without organic fertilizer were designated as CK (control), and the amount of organic fertilizer applied to the fertilized plot was 7.875 kg. The gramineous grass *Elymus nutans* ‘Aba’ was provided by Sichuan Ecological Technology Co., Ltd. The organic fertilizer (organic matter content ≥45%, N + P_2_O_5_ + K_2_O ≥ 5%, pH value 5.5–8.5) was provided by The School of Resources, Sichuan Agricultural University, which provides technical formulas mainly composed of organic-rich crop straw and poultry manure as raw materials with the addition of a biological starter.

### 2.2. Vegetation Measurements and Rhizosphere Soil Sampling

Ten plants were randomly selected and marked from each plot; the distance from the bottom of the stem to the tip of the topmost leaf or the top of the inflorescence was measured with a straightedge from the emergence of the seedlings, and the average value was calculated for each plot. In each plot where the density was uniform and the growth was average, a 50 cm × 50 cm quadrat was selected for sampling. After the plot was selected, first, the aboveground biomass was removed, and then a trowel was used to dig out the root system to a depth of 30 cm, which was then placed in a double layer of gauze to wash and remove the soil. The root system was then placed into a graduated cylinder with a specific amount of water, until the root sample was submerged; it was then gently stirred with a glass rod to remove air bubbles. After allowing to stand for a few minutes, the volume of the solution was measured and used to calculate the volume of the root system. Afterward, the root sample was taken back to the laboratory and oven-dried at 105 °C for 24 h, before weighing after cooling. The gramineous roots were washed three times with PBS buffer after bulk soils were shaken off from the roots. The suspensions were pooled and centrifuged; the resulting sediment pellets were identified as rhizosphere soils. The plant roots were submerged in 100 mL of PBS (0.02 M, pH 7.0, containing 0.1% Tween-80), and sonicated for 10 min at 20,000 Hz in an ultrasonic cleaner water bath. Then, they were freeze-dried with a vacuum drying pump, finally obtaining rhizosphere soil, which was stored in a −80 °C refrigerator.

A total of 1 g of fresh root samples of host plants were taken, and the rhizosphere soil was shaken off before being gently rinsed with water. The roots were cut into segments of about 1 cm and placed in formaldehyde acetic acid–ethanol fixative (FAA) overnight. Individual root systems were placed into a small plastic dish and moistened with dH_2_O; they were kept moist at all times. Once cleaned, roots were stored in an airtight vessel submerged in 70% ethanol. Preparation for Trypan blue staining was as follows: (1) the water bath was preheated to 100 °C, with the lid closed to accelerate heating and to avoid evaporation; (2) roots were transferred to labeled microcentrifuge tubes (2–3 roots per tube) in a microcentrifuge tube rack; (3) approximately 1 mL of 10% KOH was introduced into the microcentrifuge tubes until the roots were submerged; (4) the sample tubes were placed in a boiling bath for 20 min; (5) the KOH solution was removed from the root samples using a pipette, the roots were then rinsed three times with autoclaved or distilled water, and the staining solution was added to the samples; (6) the samples were placed back into the water bath at 100 °C for 3 min; (7) the roots were then transferred to the washing tray using blunt forceps, and approximately 1 mL of de-staining solution 1 (1% acetic acid) was added to each well to remove excess dye; (8) the roots were left to de-stain overnight; (9) de-staining solution 1 was replaced with enough de-staining solution 2 (50% glycerol) to cover the roots [32]. After the staining was completed, 10 root segments were randomly selected to complete the photography. Procedure was carried out in triplicate for each sample. The finished pressed sections were observed by light microscopy for spores, hyphae, vesicles, and other mycorrhizal structures, and those with better staining quality were photographed under fluorescence microscopy. The mycorrhizal infestation rate was calculated according to the method by Blažková et al. [33].

### 2.3. DNA Extraction and PCR Amplification

Total DNA was extracted from a 0.25 g sample of rhizosphere soil using a Power Soil™ DNA Isolation Kit (MO BIO Laboratories, Carlsbad, CA, USA) according to the manufacturer’s instructions [34]. The specific experimental operation refers to the method of [4]. Extracted DNA was amplified using the 16S rRNA universal primer set, 515 forward (5′-GTGCCAGCMGCCGCGGTAA-3′) and 806 reverse (5′-GGACTACHVGGGTWTCTAAT-3′), targeting the V4 hypervariable regions of the prokaryotic 16S rRNA genes, and supplemented with sample-specific barcodes at both 5′ ends [35]. Polymerase chain reaction (PCR) amplification was performed in a 50 μL reaction system containing 0.5 μL of Taq DNAase (TaKaRa), 1.5 μL of dNTP mixture, 1.5 μL of 10 μM forward and reverse primers, 2 μL of template DNA (10–30 ng), 5 μL of 10× PCR buffer, and 38 μL of ddH_2_O. Thermal cycling conditions were as follows: 94 °C for 1 min, 35 cycles, 94 °C for 20 s, 57 °C for 25 s, 72 °C for 45 s, and 72 °C extended for 10 min. PCR products were purified by gel electrophoresis and quantified by Qubit fluorimeter (Invitrogen, Carlsbad, CA), and then all samples were sent to Magigene Biotechnology (Guangzhou, China) to the Illumina NovaSeq platform for sequencing with a 2 × 250 bp sequencing kit. Fungal genomic DNA was extracted using the Fast DNA spin kit for soil (MP Biomedical LLC, USA) according to the manufacturer’s structure. gITS7 (5′-GTGARTCATCGARTCTTTG-3′) and ITS4 (5′-TCCTCCGCTTATTGATATGC-3′) were used as primers to amplify the ITS region [36]. The primers contain unique barcodes to allow sequencing of multiple samples in a single sequencing session.

### 2.4. Processing of the Sequencing Data

Raw 16S rRNA and ITS gene sequences were demultiplexed using an established sequence analysis pipeline (http://mem.rcees.ac.cn:8080, accessed on 30 April 2022) [35] integrated with various bioinformatics tools. Raw reads were assigned to samples on the basis of barcodes, allowing for one mismatch. Forward and reverse sequences were combined by FLASH [37]. UPARSE was used to remove chimeras and to generate OTU (operational taxonomic unit) table at 97% similarity level. Unclassified OTUs were blasted on NCBI, and plant-associated sequences were removed. The OTU table was randomly resampled with the same sequence number (5205 reads) so that downstream analyses were conducted at the same sequence depth. Sequencing data are available in the NCBI sequence read archive [38]; the minimum length of 16S was 140 bp and the minimum length of ITS was 300 bp. The resampled ASV table was used for subsequent community analysis.

### 2.5. Statistical Analysis

One-way ANOVA was used test the significance of the difference in pasture biomass and rhizosphere microbial alpha diversity (*p* < 0.05). Microbial community diversity was expressed by the Shannon–Wiener diversity index, Simpson index, and Pielou index, while principal component analysis (PCA), a linear feature vector-based unconstrained ranking method, was used to examine microbial community structure. We used the proposed normalized stochasticity ratio (NST) to quantify the significance of the difference between the microbial communities observed in CK and organic fertilizers and the random expectation of all communities. The relative importance of deterministic and stochastic processes in the construction of different communities was quantified by comparing the distribution of NST values between sample pairs within different groups: the range of NST values [0, 1] was taken with 0.5 (50%) as the cutoff point, and, if the NST of a group of communities was mainly distributed above 0.5 (50%), then stochastic processes were considered to dominate within that group of communities, and vice versa. Microsoft Excel 2010 software and SPSS 23.0 software were used for statistical analysis of data. The null model analysis was performed through an online analysis platform (http://mem.rcees.ac.cn:8080, 30 April 2022) [35]. Plots were created by Origin 2021 and Adobe Illustrator (AI).

## 3. Results

### 3.1. Application of Organic Fertilizer Increase the Forage Height and Biomass of Forage

The addition of organic fertilizer significantly increased the height and above-ground and below-ground biomass of the forage compared with CK, with an increase of 13 cm in plant height and 205 g and 100 g for above- and below-ground biomass, respectively (Figure 2).

### 3.2. Organic Fertilizer Treatment Changed the Diversity, Composition, and Structure of Rhizosphere Soil Microbial Community

For rhizosphere microbial α-diversity, the bacterial and fungal diversity indices are shown in Figure 3A. For bacteria, the Shannon, Pielou, and Simpson indices all showed a considerable increase in fertilized soils, with the Simpson index showing the most significant change (*p* < 0.01). However, the opposite trend was observed for fungi, which indicated that the diversity of bacterial communities was more susceptible to soil nutrients than that of fungal communities.

To investigate the changes in the composition of the rhizosphere microbial community following the application of additional organic fertilizer to *Elymus nutans* ‘Aba’, the rhizosphere bacterial and fungal communities were annotated at the genus level using Greengene and UNITE databases, respectively. We found that *Arthrobacter*, *Acinetobacter*, and *Pseudomonas* were dominant in the bacterial community. Among them, the relative abundance of *Acinetobacter* was significantly higher in CK than in the organic fertilizer treatment (Figure 3B). The analysis of rhizosphere fungi at the genus level showed that there were large differences in the distribution of different genera under two treatments. *Gibberella*, *Didymella*, and *Schizothecium* were the dominant genera, followed by *Kotlabase*, *Preussia*, and *Fusarium*. The relative abundances of *Didymella, Schizothecium*, and *Katabases* decreased in the organic fertilizer treatment. In contrast, the relative abundance of *Gibberella*, *Preussia*, and *Fusarium* increased significantly after the application of organic fertilizer.

Furthermore, we found that there were significant differences in rhizosphere bacterial and fungal community structure (Figure 3C). The results show that, for rhizosphere bacteria, the first principal component axis (PCA1) explained 73.8% of the variation and the second principal component axis (PCA2) axis explained 15.8% of the community variation in PCA analysis. For rhizosphere fungi, PCA1 explained 40.1% of the community variation, and PCA2 explained 11.8% of the community variation. Dissimilarity analysis (MRPP and ANOSIM) further proved that there were significant differences between the control and the organic fertilizer treatment (Table 1).

### 3.3. Variations of Ecological Processes in the Microbial Community Assembly under Organic Fertilizer Treatment

Null model analysis was used to disentangle the relative importance of stochastic and deterministic processes in microbial assembly under organic fertilizer treatment (Table 2). As the NST results show, for the bacterial community, the community building process, in both the control and the fertilizer treatments, was dominated by stochasticity. After using OTU abundance data from the communities of the two treatment groups to fit NCM (Figure 4A), the *R*^2^ approximation of the two groups was *R*^2^_CK_ = 0.7 and *R*^2^_organic fertilizer_ = 0.601. Similarly, no significant difference in the niche width distribution of OTUs in the two groups was observed. For the fungal communities, the addition of organic fertilizer switched the assembly process from stochastic to deterministic (NST_CK_ = 52%, NST _organic fertilizer_ = 48%). The DCM results showed that the organic fertilizer treatment reduced not only the stochastic process (*R*^2^_CK_ = 0.535, *R*^2^_organic fertilizer_ = 0.326) but also the ecological niche width of the fungal community compared to CK (Figure 4B).

### 3.4. Effects of Fertilizer Treatments on Rhizosphere Microbial Biomarkers

We used LEfSe (linear discriminant analysis (LDA) effect size) analysis, which enables the characterization of microbial taxa specific to an experimental or environmental condition and identifies metagenomics biomarkers in different microbial communities to identify high-dimensional biomarkers within the organic fertilizer treatment. The LDA distributions of differential functions are shown in Figure 5. The LDA analysis of the bacterial and fungal communities detected 25 (CK = 5, organic fertilizer = 20) biometrically significant biomarkers. For the bacteria communities, the higher-scoring biomarkers of the organic fertilizer treatment belonged to *Chryseolinea*, Bacteroidetes, Ferritrophicaceae, Caldilineales, Phycisphaerales, and *Litorilinea*. Meanwhile, in Fungi, the higher-scoring biomarkers of the organic fertilizer treatment belonged to Rhizophydiomycetes, *Veronaea*, Eremomycetaceae, *Cladosporium*.

### 3.5. Validation of the Effect of Organic Fertilizers on the Production of Mycorrhizal Fungi Using Arbuscular Mycorrhizal Fungi (AMF)

LEfSe results found that most of the high-dimensional biomarkers under organic fertilizer treatment were symbiont mycorrhizal fungi. In order to understand more clearly the effect of organic fertilizer treatment on mycorrhizal fungi, we observed the staining status of AMF by trypan blue staining and calculated the staining rate. The results showed that the structure of vesicles, hyphae, and tufts of AMF in the root cortex could be clearly observed by trypan blue staining. The addition of organic fertilizer increased the vesicle structure of AMF, and the number of AMF also increased significantly (*p* < 0.05). By calculating the infection rate of AMF, we found that it was significantly higher under organic fertilizer treatment (47%) than under the control CK (15%).

## 4. Discussion

The results of this experiment confirm that the application of organic fertilizer in the Qinghai–Tibet Plateau can increase the plant height of forage grass and the above- and belowground biomass, as proven in the application of organic fertilizer for the promotion of plant growth [29]. The organic fertilizer selected in this study was rich in a large number of organic matter types, nitrogen, phosphorus, potassium, and other elements, which can be a good supply of nutrients required by forage plants. Recent research has pointed out that grass plants are sensitive to nitrogen fertilizer, and that, when easily available nitrogen is present, it is quickly absorbed and utilized by plants [39], with plant height and plant biomass undergoing obvious changes [30]. At the same time, the large numbers of functional bacteria in organic fertilizer also promote plant growth [40].

In this study, the diversity and composition of rhizosphere bacterial and fungal communities were determined on the basis of high-throughput sequencing of the 16S rRNA genes and ITS regions. The results showed that the application of organic fertilizer increased rhizosphere bacterial diversity, especially the Shannon, Pielou, and Simpson indices. Of these, the Simpson index changed most significantly (*p* < 0.01), but the opposite trend was observed for fungi, indicating that organic fertilizer reduced fungal diversity (Figure 3A). Bacteria are essential for maintaining soil fertility and soil ecosystem function, and they are often sensitive to organic fertilizer inputs. Tao et al. [31] used quantitative PCR (qPCR) based on the 16S rRNA gene to estimate bacterial abundance in different long-term fertilizer treatments, and the results showed no significant difference between the NPK treatment and the unfertilized control, but the application of organic fertilizer resulted in a significant increase in bacterial abundance compared to the control. Ruibo et al. [41] found similar results, noting that bacterial abundance was much higher when NPK was added to livestock manure than when NPK was added to wheat straw. The consensus regarding the response of bacterial diversity to fertilizer treatment is that long-term NPK fertilization results in a decrease in soil bacterial abundance and diversity, while organic fertilizer treatment results in higher bacterial abundance and diversity that could even be significantly higher than NPK treatment and non-fertilizer control [42]. The effect of organic fertilizer application on the composition of the soil fungal community is mainly achieved by altering soil properties, especially that of the soil carbon pool [43]. Nutrients in organic fertilizer are more selective for fungal taxa [44], thus enhancing the effect of ecological niche filtering on fungal communities. The results of our experiments found that the application of organic fertilizer promoted species extinction and, thus, reduced fungal abundance. Taxonomic analysis showed that *Arthrobacter*, *Acinetobacter*, and *Pseudomonas* were dominant in all samples. The relative abundance of *Acinetobacter* was significantly (*p* < 0.05) higher in CK than in the organic fertilizer treatment (Figure 3B). Numerous studies have confirmed that *Acinetobacter* is a pathogenic genus that may cause varying degrees of damage to plants and humans [45]. This experiment demonstrates that organic fertilizer can inhibit the occurrence of *Acinetobacter* and, thus, improve the microbial habitats in soil. The analysis of rhizosphere fungi at the genus level showed that there were large differences in the distribution of different genera of rhizosphere fungi under different treatments. *Gibberella, Didymella*, and *Schizothecium* were the dominant genera. The relative abundances of *Didymella* and *Schizothecium* decreased in the organic fertilizer treatment. In contrast, the relative abundance of *Gibberella, Preussia*, and *Fusarium* increased significantly after the application of organic fertilizer (Figure 3B). *Gibberella* can secrete gibberellins, which promote the growth of plant stems; the genus *Didymella*, of the Ascomycota, includes potential plant parasites [46]. As an important microbial resource, *Preussia* can survive in plants without harming them, and numerous studies have found that members of this genus can significantly increase the ability of plants to withstand salt stress [47]. In the present study, the addition of organic fertilizer significantly increased the relative abundance of *Preussia*. According to our PCA results, the structure of the rhizosphere bacterial and fungal communities changed predictably under organic fertilizer treatment, which resulted in significant changes in the structural composition of the microbial community.

Application of organic fertilizer not only significantly affected the rhizosphere microbial diversity and community composition, but also significantly affected the composition pattern of rhizosphere microbial community. In this paper, the combinational patterns of rhizosphere bacterial and fungal communities were analyzed by NST and NCM, and the relative importance of deterministic and stochastic processes in the construction of different communities was quantified [48]. In this study, we found that the rhizosphere bacterial community of herbage was dominated by random process, while the rhizosphere fungal community showed a trend of transition from randomness to determinism. The relative importance of deterministic and stochastic processes in the construction of different communities was quantified by comparing the numerical distribution of NST between sample pairs within different groups [48]. If the NST of a group of communities is predominantly distributed above 0.5 (50%), then it is assumed that stochastic processes dominate within that group of communities; conversely, if the NST of a group of communities is predominantly distributed below 0.5 (50%), then it is assumed that deterministic processes dominate within that group of communities [22]. The NST results showed that the community building process of the bacterial community was predominantly stochastic for both the control and fertilized treatments (NST_CK_ = 67%, NST_organic fertilizer_ = 55%). Unlike the bacterial community, the community building process of the fungal community was predominantly stochastic in the control treatment and deterministic in the fertilizer treatment (NST_CK_ = 52%, NST _organic fertilizer_ = 48%). From these results, it can be seen that the effect of deterministic processes on bacteria was much weaker than on fungi (Table 2). To confirm this, we combined NCM (comparing model fit *R*^2^, and mobility m-values) and calculated the eco-niche width of each OTU in each group of communities. After fitting the NCM with OTU abundance data of the two groups, it was found that the influence of the diffusion restrictions of the two groups was almost identical. Fungal communities in alpine regions are more sensitive to the external environment than bacterial communities, and the microbial assembly process of fungal communities is easily disturbed by the external environment. Wang et al. [4] found that grass degradation can also cause the microbial assembly process of fungal communities to change from stochastic to deterministic. Grassland degradation would significantly reduce soil carbon and nitrogen contents [4]. In our study, the application of organic fertilizer significantly increased the nitrogen content in soil. These studies indicated that soil fungi were more susceptible to the influence of nitrogen elements in soil and, thus, changed their microbial community structure. Therefore, it is concluded that the addition of organic fertilizers in alpine areas can cause changes in microbial assembly process, especially the transformation of microbial assembly process from random to deterministic in fungal communities.

In order to further clarify the pattern of changes in bacteria and fungi in the ecological process of microbial assembly caused by organic fertilizer, we used LEfSe analysis to discover high-dimensional biomarkers [49] within the organic fertilizer treatment. For bacteria communities, the higher-scoring biomarkers of organic fertilizer treatment belonged to *Chryseolinea*, Bacteroidetes, Cytophagia, Ferritrophicaceae, Caldilineales, Phycisphaerales, and *Litorilinea*, while those for CK belonged to *Aggregicoccus*. It can be seen that the addition of organic fertilizer significantly increased the biomarkers of the bacteria community, especially in the Bacteroidetes, which are abundant pathogen-suppressing members of the plant microbiome that contribute prominently to rhizosphere phosphorus mobilization, a frequent growth-limiting nutrient in this niche [50]. The addition of organic fertilizer can help plant roots absorb nutrients by increasing this bacterial group. In fungi, the higher-scoring biomarkers of organic fertilizer treatment belonged to Rhizophydiomycetes, *Veronaea*, Eremomycetaceae, *Cladosporium*, and *Arthrographis*. The analysis revealed that the addition of organic fertilizer significantly increased the biomarkers of symbiotic fungi in the fungal community, in agreement with the NST analysis results. This indicated that organic fertilizer shifted the process of microbial assembly of the fungal community from stochastic to deterministic, and that there was a high probability of changes if the symbiont was found in the LEfSe results. In order to further clarify whether the obvious changes during microbial assembly of fungal communities in the organic fertilizer treatment were due to symbiont, we used AMF as the subject of our study to explore the effect of organic fertilizer on this group of fungi. Overall, we showed that manure organic fertilizer application increased the biomasses of AMF (Figure 6). In particular, organic fertilizer may promote AMF root colonization by enhancing AMF spore germination and growth rates. Organic fertilization elevates AMF abundance and shapes the community, which may facilitate symbiosis between AMFs and plants [51]. The presence of mycorrhiza may favor plant nutrient acquisition by speeding up the transformation of organic manure [52]. It has been found that organic fertilizers can increase the number and change the community structure of AMFs in comparison to inorganic fertilizers, and that long-term application of inorganic fertilizer can lead to a decrease in the number and activity of AMFs [53]. It has also been suggested that AMFs can help soil saprophytes decompose organic matter to increase soil nutrient levels, after which AMFs rapidly transfer nutrients from decomposing organic matter to plants [54]. In contrast to chemical fertilizers, organic fertilizers require conversion and degradation before they can be taken up by plants. Plants can rely on AMFs to facilitate the conversion process of organic fertilizer [55], which may increase their allocation of subsurface carbon to increase the number of AMFs after organic fertilizer application. In conclusion, our results point to the fact that organic fertilizers can significantly increase the rate of AMF infiltration and can promote the development of AMF vesicles, hyphae, and other structures.

## 5. Conclusions

Our study revealed that organic fertilizers significantly changed the diversity and community composition of rhizosphere microbiota of gramineous grasses. In addition, fungi were more sensitive to organic fertilizer than bacteria. The addition of organic fertilizer altered the assembly mechanism of fungal communities and reduced their niche breadth. The application of organic fertilizer could significantly increase the number and activity of AMFs. All these changes from rhizosphere microbiota could benefit the growth of gramineous grasses, which might be a promising approach to enhance the pasture yields on the Qinghai–Tibet Plateau.

## Figures and Tables

**Figure 1 microorganisms-10-01148-f001:**
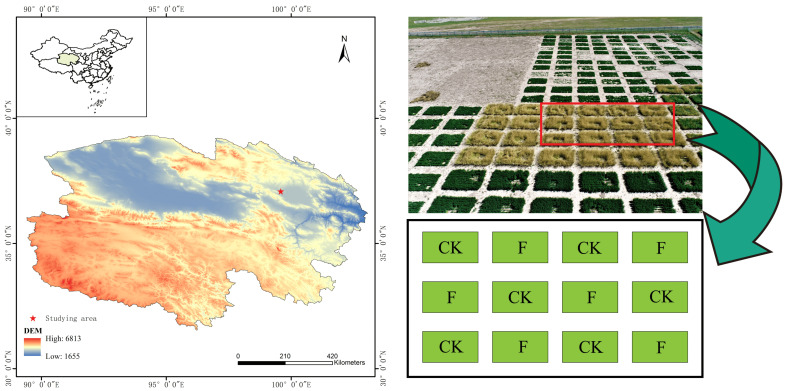
Distribution of sampling plots in the study area.

**Figure 2 microorganisms-10-01148-f002:**
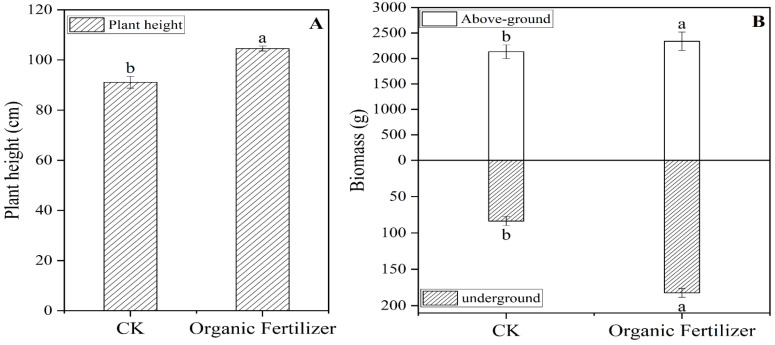
*Elymus nutans* ‘Aba’ plant height and biomass: (**A**) plant height; (**B**) biomass. Lowercase letters indicate statistically significant differences.

**Figure 3 microorganisms-10-01148-f003:**
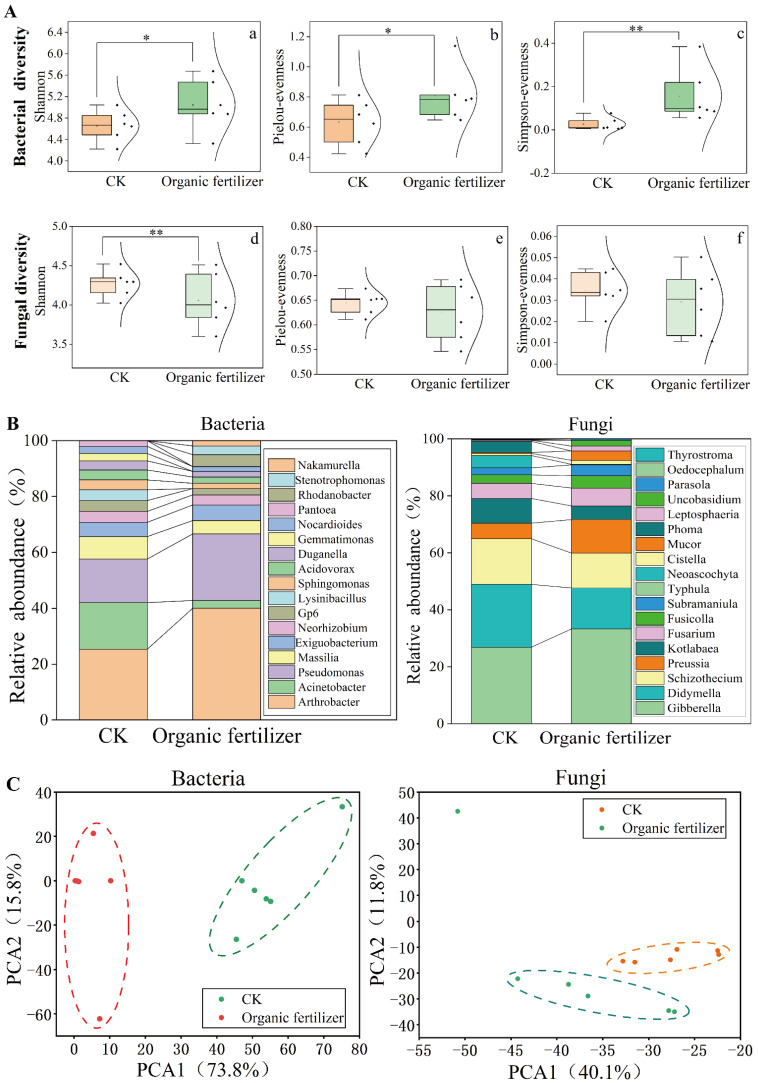
The diversity, structure, and composition of rhizosphere microbial communities in CK and organic fertilizer: (**A**) rhizosphere microbial community diversity and *t*-test (* *p* < 0.05, ** *p* < 0.01); (**B**) relative abundances of the dominant phyla in CK and organic fertilizer; (**C**) principal component analysis (PCA) of the prokaryotic community and the fungal community.

**Figure 4 microorganisms-10-01148-f004:**
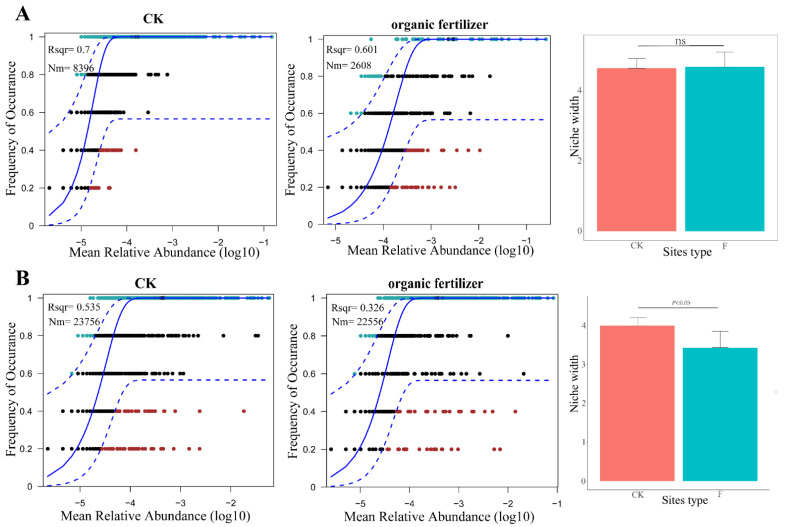
Neutral community model for bacteria (**A**) and fungi (**B**), NCM analysis. *R*^2^ represents the overall goodness of fit of the neutral community model, with higher *R*^2^ indicating that a closer model to neutral suggests that the construction of the community is more influenced by stochastic processes and less influenced by deterministic processes. Nm is the product of effective population size (N) and migration rate (m), quantifying the estimate of dispersal between communities and determining the correlation between frequency of occurrence and relative regional abundance. The solid blue line indicates the best-fit value of the neutral community model, the dashed blue line represents the 95% confidence interval of the model (estimated by 1000 bootstrap), and OTUs that occur more or less frequently than predicted by the neutral community model are shown in different colors. Eco-niche width comparison between treatments. ns indicates that the difference is not significant.

**Figure 5 microorganisms-10-01148-f005:**
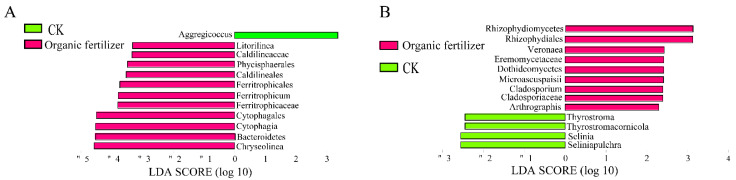
Distribution of LDA values for divergent taxa in bacteria (**A**) and fungi (**B**), with colors representing the corresponding treatments and the length of the bars representing the magnitude of the contribution of the divergent taxa (i.e., the LDA score), showing taxa with significant differences in abundance between groups under conditions where the LDA score was greater than the set value of 3, i.e., biomarkers with significantly higher abundance within each group than in all other groups. Green represents CK; red represents organic fertilizer treatments.

**Figure 6 microorganisms-10-01148-f006:**
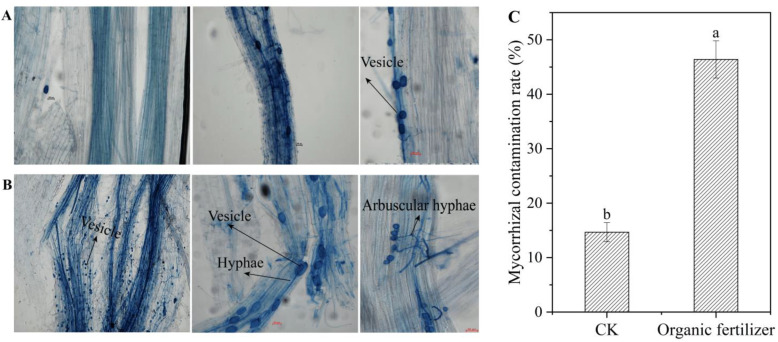
AMF infection in forage grass roots: AMF infection in control (**A**) and organic fertilizer (**B**) treatments; (**C**) AMF infection rates. Lowercase letters indicate statistically significant differences (*p* < 0.05).

**Table 1 microorganisms-10-01148-t001:** Dissimilarity test of the effect of organic matter addition on rhizosphere microbial community structure.

Dissimilarity Test	Method	Bacteria	Fungi
		Bray–Curtis	Bray–Curtis
**CK vs. organic fertilizer**	MRPP	0.6157 **	0.4222 *
	ANOSIM	0.2962 **	0.3333 *

Note: * and ** respectively represent the significance between different fertilization gradients at *p* < 0.05 and *p* < 0.01 levels.

**Table 2 microorganisms-10-01148-t002:** Effects of organic fertilizer addition on microbial community construction.

	Bacteria		Fungi	
	CK	Organic Fertilizer	CK	Organic Fertilizer
Observed dissimilarity of community	0.666 ± 0.119	0.526 ± 0.166	0.508 ± 0.094	0.472 ± 0.063
Normalized stochasticity ratio (NST)	67%	55%	52%	48%

## Data Availability

Data were deposited in the China National Microbiology Data Center (NMDC) with accession numbers SUB1651052286055. All other relevant data are available from the corresponding author on request.

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
