# Peer review of "Application of Organic Fertilizer Changes the Rhizosphere Microbial Communities of a Gramineous Grass on Qinghai–Tibet Plateau"

_microorganisms, 2022, doi:10.3390/microorganisms10061148_

Round 1

Reviewer 1 Report

Reviewer

Manuscript intitled: “Application of organic fertilizer changes the rhizosphere micro-2 bial of a gramineous grass on Qinghai-Tibet Plateau”

The authors present the effects of organic fertilizer application on the soil microbial community in grassland systems.  The effects of organic fertilizers on the structure of rhizosphere microbial communities are still limited. In this study, the diversity and composition of rhizosphere microbial communities of a gramineous grass Elymus nutans under organic fertilizer treatment have been studied in an artificial pasture on Qinghai-Tibet Plateau. The authors discovered that after a growing season, the application of organic fertilizer not only increased the height and biomass of Elymus nutans, but also changed the rhizosphere microbial compositions. In particular, organic fertilizer increased the diversity of rhizosphere bacterial community and inhibited the growth of harmful bacteria such as Acinetobacter, but the opposite trend was observed for the diversity of fungal community. The assembly process of fungal community was changed from stochastic process to deterministic process, indicating that selection had been strengthened. Additionally, both the infection rate of arbuscular mycorrhizal fungi (AMF) to host plants and the development of AMF-related structures were significantly increased after the application of organic fertilizer. Our study demonstrated that the addition of organic fertilizer to artificial pasture can improve the growth of grass through the alteration of the rhizosphere microbial communities. Organic fertilizer has a greater selectivity on both bacterial and fungal community that enhanced the niche filtration on this community, and further benefited the yield of forages.

The presented manuscript contains a study that is proper for Microorganisms Journal.

The information presented in the manuscript is new, original, well-structured and documented.

  • The title reflects the content of the manuscript.
  • The manuscript is sustained by a suitable literature. The manuscript is clearly presented!
  • The introduction offers the proper arguments for the objectives of the study.
  • The manuscript hypothesis is clear presented.
  • The methodology chapter is very detailed described. The authors use modern methods (as DNA extraction and PCR amplification).
  • The used statistical analysis is correctly used and proper for the described study subject.
  • The results and discussions are proper described and presented. The authors present new and original ecological data.
  • The tables and figures reveal properly the described results/data.
  • References:

Please, check if the all references were found in the manuscript and vice versa.

Please, follow the instruction for the authors for references.

Check the English language again with a native English speaker!

On the other hand some minor comments must be added:

  1. INTRODUCTION

I strongly recommend specifying that are similar studies in Asia (especially) or in the world? What they demonstrated? What brings new and original this study! Only the fact that the study is made in Qinghai-Tibetan plateau?

  1. MATERIALS and METHODS.

Please, insert the name of Statistical Software! Adobe Illustrator is software for graphic design! Some methods are not described here, as: null model analysis, linear discriminant analysis.

Lines 139-143. Please, specify if those data are original and their own data of the authors. If not, please insert references!

  1. DISCUSSIONS

But at discussion chapter, the author does not mention if there are similar studied in Asia or in world?  If they are, a comparison between data is neccesary…especially with those from Asian countries!

  1. FIGURES: I recommend to make more visible the figures 3 A and B. It very difficult to see especially the relative abundance at bacteria and fungi genera! The same situation is on Figure 5.

  1. AUTHOR CONTRIBUTIONS: There are some neconcordance regarding the names and the number of contributed authors! Please, see the manuscript!

Please, see the minor spelling corrections from the manuscript.

All comments were inserting in the manuscript!

Author Response

Dear Editor,

Thank you very much for your thoughtful comments and considerate suggestions for our manuscript. These comments are valuable and helpful for improving our manuscript. We have made careful modifications and revisions on the original manuscript in response to the reviewers’ comments. We hope the new version of the revised manuscript would meet the microorganism’s standard. Answers to referee’s questions are blue. We greatly appreciate the editor and reviewers’ work and valuable suggestions during this hard period.

Replies to Reviewer #1:

1) * INTRODUCTION

I strongly recommend specifying that are similar studies in Asia (especially) or in the world? What they demonstrated? What brings new and original this study! Only the fact that the study is made in Qinghai-Tibetan plateau?

Reply: Thanks for your suggestion. I have made improvements in response to the comments given by the experts. The effect of organic fertilizers on rhizosphere microorganisms is generally widely studied in agroecosystems in the Asian context, with few studies in alpine rangelands. (L125)

2) * MATERIALS and METHODS.

Please, insert the name of Statistical Software! Adobe Illustrator is software for graphic design! Some methods are not described here, as: null model analysis, linear discriminant analysis.

Lines 139-143. Please, specify if those data are original and their own data of the authors. If not, please insert references!

Reply: Thanks for your suggestion. I have made improvements in response to the comments given by the experts. (L254) (L135)

3) * FIGURES: I recommend to make more visible the figures 3 A and B. It very difficult to see especially the relative abundance at bacteria and fungi genera! The same situation is on Figure 5.

Reply: Thank you very much for your comments. We have remade the picture and the resolution has been increased to 600 dpi.

4) * AUTHOR CONTRIBUTIONS: There are some neconcordance regarding the names and the number of contributed authors! Please, see the manuscript!

Reply: Thanks for your suggestion. I have revised accordingly to address the comments given by the experts. (L499)

Reviewer 2 Report

The manuscript presented studies concerning the complex analysis of changes in the rhizosphere microbial community of gramineous grass after the application of organic fertilizer. The study is very interesting and contains many valuable results. The manuscript is generally well designed, well structured, and written. However several parts have to be improved before accepting. Here are some specific comments:

  1. Abstract section, line 22 “harmful bacteria” – I suggest specifying e.g.: pathogenic bacteria
  2. Introduction, line 109 – please define the abbreviation AMF. 20.11.2019 Generally abbreviations should be defined at first mention in each of the main sections of your paper.
  3. Methods and materials section, line 185. I suggest adding the subsection concerning the preparation and staining of the roots. 
  4. Line 196 – Kowal et al. 2020 – please add it to the references and cite it as a number in bracket [x].  
  5. 2.3. DNA extraction and PCR amplification section – please give more detail about the sequencing chemistry and library preparation. 
  6. 2.4. Processing f the sequencing data section – there is no information about the clustering and reference databases used. Please specify. 
  7. Line 220 “ANONA” – did you mean “ANOVA”?
  8. Line 254-256 – please specify the version of databases used
  9. Please italicize the names of taxa.
  10. Figure 3, Figure 4, and Figure 5 are unreadable. 
  11. I highly recommend that authors should grammar and spell-check the paper. 

Author Response

Revision Notes for microorganisms-1726230

Dear Editor,

Thank you very much for your thoughtful comments and considerate suggestions for our manuscript. These comments are valuable and helpful for improving our manuscript. We have made careful modifications and revisions on the original manuscript in response to the reviewers’ comments. We hope the new version of the revised manuscript would meet the microorganism’s standard. Answers to referee’s questions are blue. We greatly appreciate the editor and reviewers’ work and valuable suggestions during this hard period.

Replies to Reviewer #2:

1) * Abstract section, line 22 “harmful bacteria” – I suggest specifying e.g.: pathogenic bacteria

Reply: Thanks for your suggestion. I have made the appropriate changes (L22)

2) * Introduction, line 109 – please define the abbreviation AMF. 20.11.2019 Generally abbreviations should be defined at first mention in each of the main sections of your paper.

Reply: Thank you very much for your comments, I have made the appropriate changes. (109)

3) * Methods and materials section, line 185. I suggest adding the subsection concerning the preparation and staining of the roots.

Reply: Thanks for your suggestion. We have added subsections on root preparation and staining. (L183)

4) * Line 196 – Kowal et al. 2020 – please add it to the references and cite it as a number in bracket [x].

Reply: Thanks for your suggestion. I have made the appropriate changes. (L199)

5) * 2.3. DNA extraction and PCR amplification section – please give more detail about the sequencing chemistry and library preparation.

Reply: Thanks for your suggestion. We have added detail about the sequencing chemistry and library preparation. (L218)

6) * 2.4. Processing f the sequencing data section – there is no information about the clustering and reference databases used. Please specify. 

Reply: Thanks for your suggestion. We have made the appropriate changes. (L201)

7) * Line 220 “ANONA” – did you mean “ANOVA”

Reply: Thanks for your suggestion. We have made the appropriate changes to “ANOVA”. (L240)

8) *Please italicize the names of taxa.

Reply: Thank you very much for your comments, We have italicized the names of the taxa.

9) * Figure 3, Figure 4, and Figure 5 are unreadable.

Reply: Thanks for your suggestion. We have remade the picture and the resolution has been increased to 600 dpi.

Round 2

Reviewer 1 Report

Please, see some minor comments from manuscript!

Author Response

Revision Notes for microorganisms-1726230

Dear Editor,

Thank you very much for your thoughtful comments and considerate suggestions for our manuscript. These comments are valuable and helpful for improving our manuscript. We have made careful modifications and revisions on the original manuscript in response to the reviewers’ comments.

1) * Please, include references regarding these researchers and in short their obtained results!

Reply: Thanks for your suggestion. We have supplemented the experimental results obtained from the relevant studies in the corresponding positions. (L125-L132)

2) *If they are the author's data, please insert the standard error of the parameters!

Reply: Thanks for your suggestion. I am sorry for the confusion caused by my own words, but the weather data was obtained from the nearest weather station, as we have indicated in the text. (L146)

3) * Have been introduced at references?

Reply: Thank you very much for your comments. We have cited the relevant references in the text. (L260)
